# Influence of the Dispersion Medium and Cryoprotectants on the Physico-Chemical Features of Gliadin- and Zein-Based Nanoparticles

**DOI:** 10.3390/pharmaceutics14020332

**Published:** 2022-01-30

**Authors:** Silvia Voci, Agnese Gagliardi, Maria Cristina Salvatici, Massimo Fresta, Donato Cosco

**Affiliations:** 1Department of Health Sciences, University “Magna Græcia” of Catanzaro, Campus Universitario “S. Venuta”, 88100 Catanzaro, Italy; silvia.voci@studenti.unicz.it (S.V.); gagliardi@unicz.it (A.G.); fresta@unicz.it (M.F.); 2Electron Microscopy Centre (Ce.M.E.), Institute of Chemistry of Organometallic Compounds (ICCOM), National Research Council (CNR), 50019 Sesto Fiorentino, Italy; salvatici@ceme.fi.cnr.it

**Keywords:** freeze-drying, gliadin, nanoparticles, proteins, stability, zein

## Abstract

The evaluation of the physico-chemical features of nanocarriers is fundamental because the modulation of these parameters can influence their biological and in vivo fate. This work investigated the feasibility of saline, 5% *w*/*v* glucose and phosphate-buffered saline solution, as polar media for the development of nanoparticles made up of two vegetal proteins, zein from corn and gliadin from wheat, respectively. The physico-chemical features of the various systems were evaluated using dynamic and multiple light scattering techniques, and the results demonstrate that the 5% *w*/*v* glucose solution is a feasible medium to be used for their development. Moreover, the best formulations were characterized by the aforementioned techniques following the freeze-drying procedure. The aggregation of the zein nanoparticles prepared in water or glucose solution was prevented by using various cryoprotectants. Mannose confirmed its crucial role in the cryopreservation of the gliadin nanosystems prepared in both water and glucose solution. Sucrose and glucose emerged as additional useful excipients when they were added to gliadin nanoparticles prepared in a 5% glucose solution. Specifically, their protective effect was in the following order: mannose > sucrose > glucose. The results obtained when using specific aqueous media and cryoprotectants permitted us to develop stable zein or gliadin nanoparticles as suspension or freeze-dried formulations.

## 1. Introduction

The implementation of nanoparticles in biopharmaceutical applications has witnessed enormous growth as a consequence of the possibility of modulating the pharmacological properties of various bioactives [1]. In this context, the development of formulations made up of natural-based raw materials furnishes several advantages due to their biocompatibility, biodegradability, and the opportunity of using non-hazardous solvents in their preparation [2,3].

The large intrinsic surface-area-to-volume ratio of nanocarriers and the properties of the dispersant medium, as well as the pH, all influence their steric or electrostatic stabilization. The modulation of these parameters may furnish feasible approaches for preventing their aggregation [4,5]. In this regard, Tseng et al. investigated the influence of ethanol and water used as models of organic and inorganic dispersing media, respectively, over the physico-chemical features of gold nanoparticles [6]. They demonstrated that the solvent used in the preparation procedure of the carriers had a great impact on their features, especially regarding their shape. Namely, in water, they appear as oval-shaped aggregates, whereas their form is closer to spherical when ethanol is used as the dispersing medium. Unfortunately, the exact mechanism of this interesting phenomenon is not yet understood [6].

Allouni and coworkers investigated the size of titanium dioxide nanoparticles in Roswell Park Memorial Institute (RPMI 1640) culture medium with and without 10% *v*/*v* of fetal bovine serum (FBS) or human serum albumin (0.01–1% *w*/*w*). It was observed that both the composition of the medium and its ionic strength influence the aggregation of the systems [7].

In another experimental investigation, Park and Lee aimed to identify a suitable dispersing medium useful for in vitro/in vivo evaluation of the toxicological profiles of silver nanoparticles [8]. The stability of the carriers was preserved when a 5% *w*/*v* glucose solution was used for the analyses, whereas saline and a phosphate buffer solution both destabilized the colloidal architecture [8]. This was because glucose is a non-electrolytic compound and is unable to influence the surface charges of the carriers [8].

Recently, Ross et al. investigated the variations of the physico-chemical features of gold, titanium, silver, and iron oxide nanostructures, using Dulbecco’s modified eagle medium as dispersant [9]. This was done to simulate the injection of the nanoparticles following intravenous administration for drug delivery [9]. Among these, the silver- and gold-based nanocarriers were the most stable as they showed no consistent variations in their diameter after 24 h incubation. Even though it is possible to recreate the composition of biological media to forecast the behavior of a formulation following administration, it should be considered that this simulation lacks the influence of the shear stress of the bloodstream, and this aspect has been shown to greatly impact the composition of the protein corona [10].

Water is the most common media used for developing colloidal formulations to be used in the pharmaceutical field. However, aqueous suspensions can be characterized by several chemical and/or physical instability phenomena such as aggregation, sedimentation, and polymer degradation [11]. The removal of water from a formulation containing nanoparticles can promote the long-term stability of the systems, together with prolonged pharmacological efficacy. In fact, the conversion of a colloidal suspension into a solid dosage form through freeze-drying favors the elimination of water through sublimation. This is to prevent the alteration of the physico-chemical and technological features of the formulation, avoiding the degradation of the particles and the entrapped compound(s) [11]. However, the process can have a negative impact on the colloidal architecture due to the formation of ice crystals and water removal, so the addition of a cryoprotectant is often required [12]. In particular, a common strategy relies on the exploitation of sugar molecules because they are chemically inert and can create a vitreous matrix able to prevent particle aggregation [13,14,15,16].

Different hypotheses have been formulated to elucidate the stabilization mechanisms provided by these excipients. For example, it has been reported that sugars can act as surrogates of the sublimated water molecules, linking the polar moieties of the polymers (water replacement theory), and this mechanism has been well documented for liposomal formulations [17]. In addition, they prevent the formation of macroaggregates which entrap the carriers within a glassy matrix that reduces their mobility and promotes the formation of clusters (vitrification theory). This is also true in the case following the disruption and rearrangement of the tetrahedral structure of water (kosmotropic effect) [18]. The latter mechanism has been well documented for the disaccharide trehalose [19,20,21,22], but it is likely that a combination of all the aforementioned mechanisms can occur upon cryodesiccation [18]. Dynamic and static multiple light scattering (DLS and SMLS, respectively) are two common, time-saving analytical techniques used to characterize a colloidal formulation [23,24,25]. In this regard, Lazzari et al. demonstrated that DLS can be exploited to evaluate the physico-chemical features of poly-lactic acid- and poly-methyl-methacrylate-based nanoparticles in different buffers, simulated biological fluids, and tissue homogenates [26]. It was discovered that these investigations should be performed before the in vivo experiments to avoid potentially detrimental parameters during the administration of the systems [26]. A similar approach was used for alginate-based, solid, lipid microparticles proposed for the lung delivery of fluticasone propionate [27] and recently, for metallic- and silica-based nanosystems [9,28]. The SMLS technique is used to obtain qualitative information concerning the stability kinetic of formulations used for biomedical, pharmaceutical, and industrial applications [29,30,31,32]. In this experimental work, both DLS and SMLS were used to investigate the influence of various biological media/buffers on the physico-chemical features of polymeric nanoparticles made up of two plant proteins, zein, and gliadin. Freeze-drying experiments were also carried out to evaluate the possibility of developing formulations characterized by long-term storage stability. This was done by investigating the role of the medium used during the sample preparation and the effect of the cryoprotectants. The idea was to furnish information concerning the behavior of the proposed protein nanoparticles as a function of the medium used for their development and for the freeze-drying procedure that will be described further on.

## 2. Materials and Methods

### 2.1. Materials

Zein, gliadin, sodium deoxycholate monohydrate (SD), 3-[4,5-dimethylthiazol-2-yl]-3,5-diphenyltetrazolium bromide salt (used for MTT tests), dimethyl sulfoxide, amphotericin B solution (250 μg/mL), glucose, and phosphate-buffered saline tablets were purchased from Sigma Aldrich (St Louis, MO, USA). Super Refined Brij O2 (SRBO2) was purchased from Croda (Snaith, UK). Ethanol was obtained from Carlo Erba SpA (Rodano, Italy).

### 2.2. Preparation and Physico-Chemical Characterization of Protein-Based Nanoparticles

Protein-based nanosystems were prepared according to the nanoprecipitation procedure.

In detail, 3.3 mg/mL of zein and 1.66 mg/mL of gliadin were solubilized in 3 mL of an ethanol/water mixture (2:1 *v*/*v*, pH 10 for gliadin), then 5 mL of MilliQ water were added. 1.25% *w*/*v* of SD was added in the aqueous phase, whereas 0.1% *w*/*v* of SRBO2 was added in the organic phase to stabilize the zein- and gliadin-based nanocarriers, respectively, as previously reported [33,34]. Each nanosuspension was homogenized for 1 min at 24,000 rpm with an Ultraturrax^®^ (model T25 IKA^®^ Werke GmbH and Co., Staufen, Germany) and stirred on a mechanical plate at 600 rpm for 8 h to promote the evaporation of the organic solvent. The final protein concentration was 2 mg/mL and 1 mg/mL for zein and gliadin nanoparticles, respectively. The nanoparticles were purified by centrifugation using Amicon^®^ ultracentrifugal filter units (cut-off 10.000 Da, 4000 rpm, 2 h) to remove any unreacted compound [35].

The influence of various media, i.e., glucose solution (5% *w*/*v*), saline solution (NaCl 0.9% *w*/*v*), and PBS (0.01 M, pH 7.4) on the development of the systems was investigated using these polar media instead of MilliQ water during the phases of preparation. The mean particle sizes polydispersity index and Z-potential of the investigated protein-based nanoparticles were evaluated by PCS at 25 °C using a Zetasizer Nano ZS (Malvern Panalytical Ltd., Spectris plc, Malvern, UK), after diluting each sample at a 1:50 ratio in different polar media and applying the third-order cumulant fitting correlation function. The results are expressed as the average of three different experiments carried out on three different batches (10 runs for each batch) ± standard deviation [36].

Moreover, the morphology of the zein and gliadin-based nanosystems prepared in the different polar media was investigated using transmission electron microscopy (TEM), as previously reported [37].

### 2.3. Evaluation of the Stability by Multiple Light Scattering

The turbiscan Lab Expert^®^ (Formulaction, Toulouse, France) is a time-saving apparatus conceived to provide information regarding the occurrence of sample alteration due to physical phenomena such as particle migration (creaming/sedimentation) or size variations (flocculation/coalescence) as a function of time and temperature [32]. Namely, during each measurement, an electroluminescent diode operating in near-infrared (λ 880 nm) runs over the total height of the glass vial. Two sensors (the transmission and backscattering detectors, respectively) receive the light transmitted (T) and backscattered (BS) from the particles contained in the sample. Following the freeze-drying procedure, the variations in the BS and T profiles of the samples were expressed as Turbiscan stability index (TSI) according to the following equation [32]:(1)$$\mathrm{TSI}=\sqrt{\frac{\sum_{\;i=1}^{\;n}\;\left(x_{i}-x_{T}\right)}{n-1}}$$
where X_i_ is the mean backscattering obtained during each minute of analysis, X_T_ is the average of X_i_, and n is the number of scans performed throughout the analysis.

### 2.4. Freeze-Drying of Zein and Gliadin Nanoparticles

Cryodesiccation of the investigated protein-based nanoparticles was performed by enriching each formulation with various cryoprotectants (glucose, mannose, sucrose, trehalose, and mannitol at 5 and 10% *w*/*v*). Namely, 1 mL of sample was placed into pyrex glass vials, frozen in liquid nitrogen for two minutes, and then placed into the drying chamber for 24 h using a Vir Tis drying system (Vir Tis SP scientific sentry 2.0, SP Industries, Warminster, PA, USA). An applied vacuum of 30–50 mT and a condenser temperature of −55 °C were maintained throughout the process, as previously reported [36,38]. At the end of the process, the freeze-dried powder was reconstituted with the same amount of sublimated water or 5% *w*/*v* glucose and analyzed by DLS [36,38].

The protective effect exerted by the various cryoprotectants was expressed as redispersibility index (RDI) that was calculated according to the following equation [39]:(2)$$\mathrm{RDI}\;\left(\%\right)=\frac{D}{D_{0}}\times 100$$
where D is the size of the samples after the freeze-drying process, whereas D_0_ is the hydrodynamic diameter before the procedure. RDI values close or equal to 100% identify samples that can be properly re-suspended, while values below or above 100% are typical of systems characterized by a poor re-dispersibility [40].

### 2.5. Cytocompatibility and Cell Interaction of Protein-Based Nanoparticles

The cytotoxicity of the formulations prepared in various polar media was investigated by MTT assay [33,38]. In detail, human colon carcinoma and human keratinocytes (CaCo-2 and NCTC-2544, respectively) were cultured in plastic dishes (100 mm × 20 mm) in a water-jacketed CO_2_ incubator (Thermo Scientific, Dreieich, Germany) at 37 °C (5% CO_2_) using a DMEM with glutamine, supplemented with penicillin (100 UI/mL), streptomycin (100 µg/mL), amphotericin B (250 µg/mL) and FBS (10% *v*/*v*). The cells were plated in 96-well culture dishes (7 × 10^3^ cells/0.2 mL) and treated with increasing concentrations of nanoparticles (1, 10, 50, 100 μg/mL of zein and 10, 25, 50, 100 μg/mL of gliadin) at different incubation times (24, 48 and 72 h) [34,41]. Untreated cells were used as control. Successively, 20 μL of tetrazolium salt previously solubilized in a PBS solution (5 mg/mL) were added to each well, and the plates were incubated again for 3 h. Successively, the plates were analyzed using a microplate spectrophotometer (xMARK™ Bio-Rad Laboratories Inc., Hercules, CA, USA) at a wavelength of 540 nm with a reference set at 690 nm.

Cell viability, expressed as a percentage, was evaluated as the mean of 3 different experiments ± standard deviation and was calculated as follows:Cell viability (%) = AbsT/AbsC × 100(3)
in which AbsT is the absorbance of the treated cells and AbsC is the absorbance of the untreated cells (control).

The interaction between the cells and the zein and gliadin nanosystems was investigated as a function of the incubation times, using a radiolabeled hexadecyl cholesteryl ether (^3^H-CHE) marker (0.003% *w*/*w* with respect to the amount of gliadin and zein), as previously reported [33,38].

### 2.6. Statistical Analysis

The statistical analysis of the various experiments was performed by ANOVA and the results were confirmed by a Bonferroni *t*-test, with a *p*-value < 0.05 considered statistically significant.

## 3. Results and Discussion

### 3.1. Physico-Chemical Characterization of Protein-Based Nanoparticles

Currently, the translation of nanoparticles from bench to bedside is hampered by the controversial results obtained between their in vitro and in vivo features. In particular, it would be useful to investigate their behavior not only in their “original media” (i.e., in the solvent used during the preparation procedure) but also in physiological media to evaluate their performance once administered in the body [9].

Our research team recently focused on using zein and gliadin as versatile biomaterials for obtaining nanoparticles to be employed as drug delivery systems [33,34,35,36,41,42].

In particular, two formulations characterized by suitable technological features were developed following the nanoprecipitation of 2 mg/mL of zein and 1 mg/mL of gliadin [34,41]. In detail, the use of 1.25% *w*/*v* of SD and 0.1% *w*/*v* of SRBO2 increased the stability of the zein- and gliadin-based nanosystems, respectively, which promoted the retention and controlled release of various compounds [33,34,35,41].

Considering these results, the first steps of this investigation were focused on the evaluation of the influence of various media in the preparation of the protein-based nanoformulations to promote their administration in humans; namely, 5% *w*/*v* glucose solution, 0.9 % *w*/*v* NaCl solution, and PBS (0.01 M, pH 7.4) were chosen as models of polar media solutions usually used in clinical practice or as systems characterized by suitable properties (osmolarity, pH and ion concentrations) [43,44].

As shown in Table 1, both protein-based samples prepared in MilliQ water were characterized by mean sizes of ~140 nm, a monomodal size distribution, and a negative surface charge of ~−30 mV. A consistent size increase was observed when a sodium chloride solution was used as a polar medium. This could be due to the modulation of the shear plane of the nanoparticles; indeed, the composition of the dispersing media exerts great influence on the aggregation phenomena of nanoparticles, especially for protein-based systems [45]. It is probable that the presence of Na^+^ ions neutralizes the stabilizing effect provided by the emulsifiers, promoting the aggregation of the particles, as previously reported for gliadin nanoparticles stabilized with 0.25% *w*/*v* of glutaraldehyde and 0.1% *w*/*v* of pectin [46,47]. This same trend was observed when PBS was used as an aqueous phase during the preparation of the samples (Table 1). This was due to the reduction in the thickness of the double electrical layer of the carriers resulting from the high ionic strength that characterizes this solution [48,49], as confirmed by the low surface charge values obtained (Table 1). On the contrary, a 5% *w*/*v* glucose solution induced no significant variations of the mean diameter and size distribution of the systems. This was probably a consequence of the formation of hydrogen bonds between the hydroxyl groups of the sugar and the polyoxyethylene residues of the surfactants [50,51]. These results were confirmed through TEM analysis (Figure 1), which showed that gliadin and zein-based formulations prepared in glucose solution or MilliQ water are characterized by a spherical morphology while in a PBS and NaCl solution, a significant degree of aggregation occurred (Figure 1).

The SMLS technique was used to obtain information concerning the influence of the media under investigation towards the stability of the nanoparticles previously discussed. It is possible to observe the variation of the transmission and backscattering profiles of the investigated protein-based formulations as a function of the media used for their preparation in Figure 1. In particular, a decrease of the backscattering and transmission profiles of the samples prepared in PBS and saline solution was observed, demonstrating the occurrence of coalescence phenomena [52,53] and corroborating the data previously discussed.

Contrarily, the formulations prepared in a 5% *w*/*v* glucose solution showed backscattering profiles close to those observed when they were prepared in MilliQ water (Figure 2), and this trend can be better appreciated by studying the TSI profiles of the formulations (Figure 3). In fact, the formulations prepared in PBS and NaCl demonstrated a significant variation in TSI as a function of the incubation time, confirming the appearance of a certain destabilization of the samples [54,55]. The results confirm the feasibility of using a glucose solution as a polar medium useful for developing the protein-based nanoparticles (Figure 3).

The best formulations (i.e., prepared in MilliQ water or in glucose solution) were analyzed in various polar media to evaluate the potential influence of the dispersant on their physico-chemical properties (Table 2). Moreover, in this case, the presence of salts in the dispersant compromised the stability of the samples, promoting an increase in the size distribution of the nanoparticles, especially the gliadin-based ones, which evidenced a consistent reduction of their surface charge while the use of MilliQ water or a glucose solution induced little variation/improvement of the aforementioned parameters.

### 3.2. Freeze-Drying of Protein-Based Nanoparticles

Considering the previous results, the freeze-drying experiments were carried out on the formulations prepared in water and 5% *w*/*v* glucose solution as polar media. The protective effect of two disaccharides (sucrose and trehalose) and three monosaccharides (glucose, mannitol, and mannose) commonly used in cryodesiccation protocols was investigated at 5 and 10% *w*/*v*. These concentrations have been shown to generally exert a suitable protective effect on different colloidal nanosystems [13,15,56].

As can be seen in Figure 3, the gliadin nanoparticles prepared in MilliQ water and lyophilized without the addition of any cryoprotectant were characterized by a significant increase in their mean diameter and size distribution. The addition of a cryoprotectant in amounts inferior to 10% *w*/*v* cannot provide sufficient stabilization to gliadin nanoparticles, data in agreement with other experimental reports [57,58,59]. A slight increase in the sizes of systems was also observed when 10% *w*/*v* of cryoprotectants was used, but it was not very noticeable. The surface charge showed a significant variation in some cases, demonstrating Zeta potential values of about −50 mV, and this is probably due to a structural destabilization of the protein matrix resulting from variations in the exposure of the charged moieties of the biopolymer. However, when the gliadin samples were enriched with 10% *w*/*v* of mannose, an acceptable mean diameter of systems was obtained upon rehydration in water. Corroboration of these findings emerged through RDI estimation, which evidenced an increase of ~19% in the mean sizes of the gliadin nanoparticles after the freeze-drying process when 10% *w*/*v* of mannose was used as a cryoprotectant (Figure 4, Table 3). Interestingly, the worst results were obtained using trehalose (Figure 4, Table 3).

The freeze-dried zein nanosystems prepared in distilled water were characterized by a very low variation of their sizes and also by a monodispersed population, except for the sample that was freeze-dried without the use of cryoprotectants (Figure 4). Moreover, in this case, the RDI values highlighted the efficiency of their reconstitution. In particular, the nanoparticles were characterized by values of 100% in the cases of the majority of the cryoprotectants used, suggesting that zein nanostructures can be successfully redispersed in water (Table 3). This hypothesis was supported by zeta potential measurements, which showed little variations upon rehydration compared to the pristine formulation. Indeed, the surface charge values were ~−40 mV for most of the excipients analyzed, confirming that stable freeze-dried nanoformulations had been made. However, problems did occur in the presence of 10% *w*/*v* of both glucose and particularly mannose because the formation of a significant amount of aggregates was observed, and this was due to Z-potential values close to neutrality (Figure 4, Table 2). These results can be attributed to the different chemical structures of the two sugars; in fact, the spatial conformation of polyalcohols greatly influences their stabilization rate with respect to the frozen mass [60].

In particular, since mannose is the C-2 epimer of glucose, its structure comprises a reduced number of hydroxyl (-OH) moieties oriented on the same side of the carbon chain (four and five -OH residues occur in mannose and glucose, respectively). The decreased number of -OH groups that interact with the polymeric matrix compromises their protection rate [61]. The same trend has been observed for poly-lactide-co-glycolic acid-based nanoparticles enriched with lactose and glucose [62].

It is interesting to observe the results obtained in the presence of the non-reducing sugars trehalose, mannitol, and sucrose. Namely, their use as cryoprotectants endows several advantages such as Maillard browning reactions, particularly detrimental to protein-based nanoformulations, as recently demonstrated for Brij O10-stabilized zein nanoparticles [36]. Moreover, the absence of internal hydrogen bonds enhances the interaction of these cryoprotectants with the carriers, as is true for other nanosystems such as chitosan- [63,64] and zein-based nanoparticles proposed for the oral delivery of dapoxetine and resveratrol [65,66]. In fact, trehalose and sucrose have already demonstrated themselves to be particularly suitable for stabilizing protein-based nanoparticles made up of human serum albumin [67].

As shown in Figure 3, the freeze-dried zein-based nanoparticles remained stable even when prepared in the glucose solution, their surface charge being scarcely affected by the change of the polar media. Only the nanoformulation enriched with 10% *w*/*v* of glucose evidenced the formation of a population characterized by an increased polydispersity index (up to 0.30).

Gliadin-based nanoformulations prepared in the glucose solution confirmed the trend previously observed when MilliQ water was used as a polar medium during the sample preparation: indeed, both mean sizes and PdI values remained acceptable when 10 *w*/*v* of mannose was used, while the other cryoprotectants exerted a negative influence on the mean sizes of the nanosystems (Figure 4, Table 3) In fact except for the mannose, most of the samples showed a great tendency towards aggregation, as confirmed by Z-potential evaluation.

Unfavorable physico-chemical features were obtained following the lyophilization of the cryoprotectant-free formulation, demonstrating the insufficient contribution of the glucose molecules contained in the preparation medium during the sample preparation. This evidence suggests that adding a cryoprotectant is a mandatory step to manage a powder-like gliadin formulation.

Considering the results, the sizes and the PdI of the freeze-dried gliadin- and zein-based nanoparticles prepared in 5% *w*/*v* glucose solution as an aqueous phase were also investigated as a function of the media used to perform the DLS analysis (Figure 5). In detail, a comparison between the freeze-dried samples incubated in MilliQ water (as previously reported in Figure 4) or in the same medium used for their preparation was performed. As shown in Figure 5, the zein nanoparticles analyzed in MilliQ water or glucose solution showed a homogenous size distribution and a mean diameter of 130–150 nm; only the sample prepared with 10% *w*/*v* of glucose as a cryoprotectant showed an increase of the value up to 0.30, probably because of the high concentration of glucose molecules. This result highlights the fact that the maximum stabilization provided by the glucose excipient was reached, and any further increase in its concentration would not be beneficial for the colloidal architecture [68], data in agreement with those observed by de Chasteigner and coworkers for freeze-dried SD-stabilized poly-ε-caprolactone nanospheres [50]. The PdI of the zein samples containing mannose, sucrose, and trehalose evidenced concentration-dependent cryoprotection; in fact, the value of this parameter decreased when the concentration of the excipients was increased (Figure 5). However, this was not true in the case of mannitol.

A different trend was observed for freeze-dried gliadin nanoparticles; in fact, the use of a 5% glucose solution as a dispersant for the analysis promoted a decrease in the mean diameter of the gliadin nanoparticles prepared in the same medium and enriched with 5% mannose, mannitol or trehalose before the cryodesiccation as compared to MilliQ water (Figure 5) even though a polydispersed population did occur. This could be related to the Z-potential values obtained after the procedure (~−10 mV), which were not useful for ensuring a suitable electrostatic repulsion of particles, evidencing the failure of these excipients as cryoprotectants. Moreover, the glucose solution induced a significant increase in the mean sizes of the samples when 10% mannose was used as cryoprotectant with respect to the analysis performed in MilliQ water, but this system was characterized by the lowest increase of PdI. The positive contribution of this excipient was confirmed by the fact that the samples enriched with mannose demonstrated the lowest surface charge value compared to the others (Figure 5).

### 3.3. In Vitro Cytotoxicity and Interaction

The cytotoxicity of the protein formulations and CaCo-2 and NCTC-2544 cell lines was evaluated as a function of the polymer concentration and incubation times (24, 48, 72 h), as previously reported [33,34,41].

The idea was to investigate the influence exerted by the excipient used in the freeze-drying procedure over the in vitro tolerability of the protein nanosystems, and this was done with respect to two models of human cancer and normal cell lines. In parallel, cell uptake studies were performed to evaluate any influence exerted by the selected excipients over the rate of interaction of SRO2- and SD-stabilized gliadin and zein nanoparticles, respectively.

Namely, considering the results previously discussed, gliadin nanoparticles enriched with 10% *w*/*v* of mannose were chosen as the most promising nanoformulation to be used for in vitro experiments. As shown in Figure 6, cryoprotectant-free nanosystems were characterized by a safe profile of up to 25 µg/mL of protein while greater concentrations promoted a decrease in cell viability, especially after 48 and 72 h incubation, supporting our previous findings [34]. A similar trend was shown for the formulation containing mannose as a cryoprotectant. In fact, significant toxicity was observed only when concentrations higher than 25 µg/mL of protein were used, demonstrating that the presence of the excipient did not significantly influence the cytocompatibility of the nanocarriers.

As previously demonstrated, cryoprotectant-free SD-stabilized zein nanoparticles were characterized by suitable safety up to a limit of 50 µg/mL of biopolymer [33,41]. The addition of trehalose as excipient revealed the absence of significant variations of this parameter; in fact, MTT assay evidenced a decrease of cell viability by approximately 25–30% only when greater concentrations of zein were tested, and this was true for both cell lines after 24 h incubation. The other cryoprotectants analyzed (i.e., glucose, sucrose, mannose, and mannitol) showed the same trend (data not shown).

The evaluation of the interaction rate between the tritiated protein-based nanosystems and the aforementioned cell lines revealed a consistent cell uptake after just 1 h, data in agreement with our previous findings [33,69]. This evidence supports the suitability of the proposed nanocarriers to be efficiently internalized by cells and their use as drug delivery systems. Once again, the presence of mannose and trehalose in gliadin and zein nanoparticles, respectively, did not affect the degree of their internalization (Figure 7C,D).

## 4. Conclusions

In this work, we evaluated the influence of different dispersing media on the physico-chemical features of two protein-based nanoparticles by using two widely-used analytical techniques, DLS and SMLS. It was demonstrated that a 5% *w*/*v* glucose solution is a suitable alternative medium to be used for the development of zein and gliadin nanoparticles, and this is a remarkable finding, considering that a glucose solution is commonly used as diluent for injectable preparations [70,71]. Freeze-drying experiments were performed to investigate the possibility of obtaining a solid dosage form of the protein-based systems that could be rehydrated as needed. The nature and concentration of the cryoprotectant used and the characteristic of the dispersant was shown to modulate the sizing of the samples, whereas no influence was exerted over their cytotoxicity or cell internalization. This data can be useful for predicting the suitable medium to be used during the preparation of gliadin- and zein-based systems containing active compounds while keeping in mind the required administration route.

## Figures and Tables

**Figure 1 pharmaceutics-14-00332-f001:**
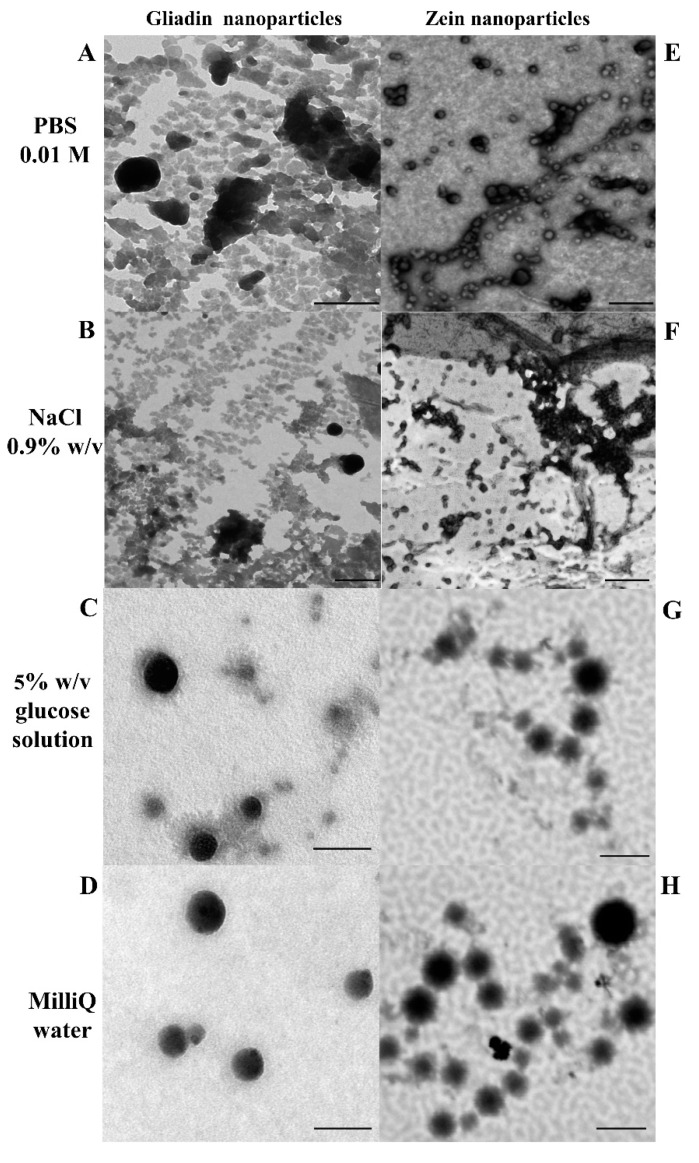
TEM images of gliadin- (1 mg/mL of protein, 0.1% *w*/*v* of SRO2) and zein-based nanoparticles (2 mg/mL of protein, 1.25 % *w*/*v* of SD) as a function of the polar media used during the preparation procedure. (**A**,**E**) bar = 1 µm. (**B**,**F**) bar = 500 nm. (**C**,**D**) bar = 200 nm. (**G**,**H**) bar = 150 nm.

**Figure 2 pharmaceutics-14-00332-f002:**
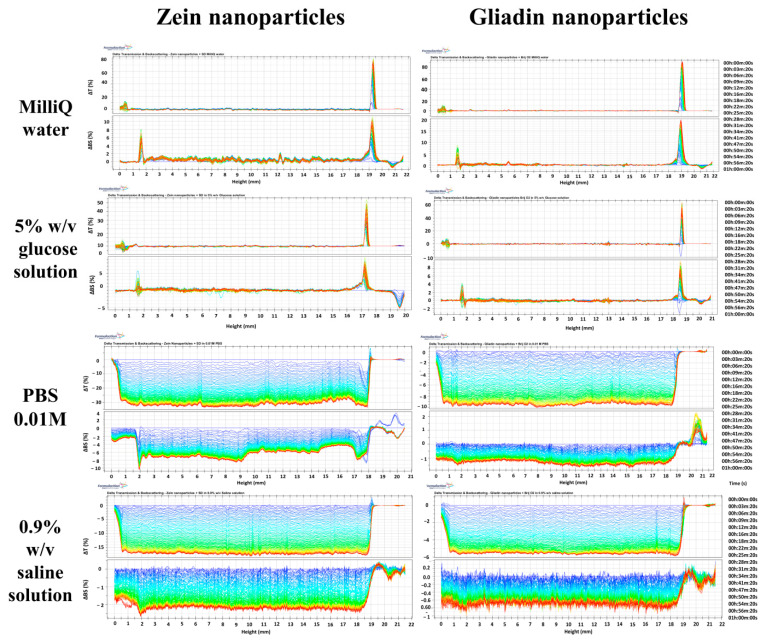
Delta Transmission (ΔT) and Backscattering (ΔB) profiles of SD-stabilized zein- and SRBO2-stabilized gliadin nanoparticles prepared in various polar media. The analysis was performed at room temperature for 1 h.

**Figure 3 pharmaceutics-14-00332-f003:**
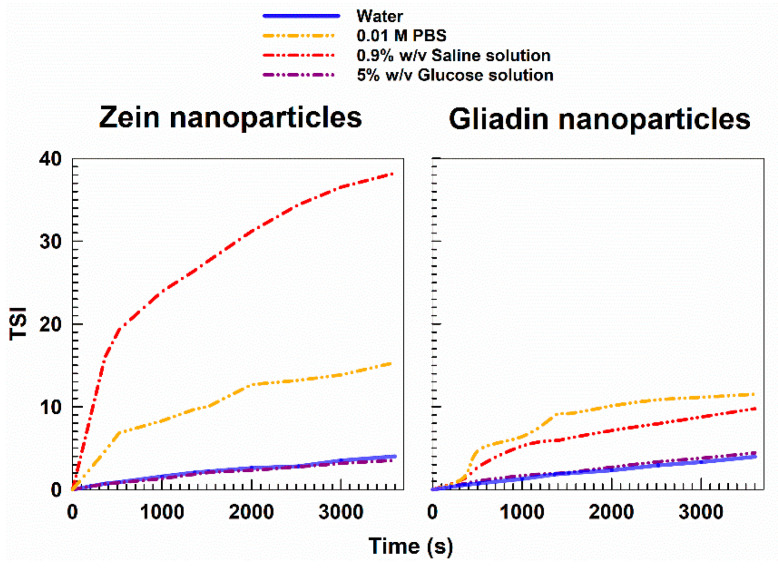
TSI (Turbiscan Stability Index) profiles of SD-stabilized zein and SRBO2-stabilized gliadin nanoparticles as a function of the medium used during the preparation procedure. The analyses were performed for 1 h at room temperature.

**Figure 4 pharmaceutics-14-00332-f004:**
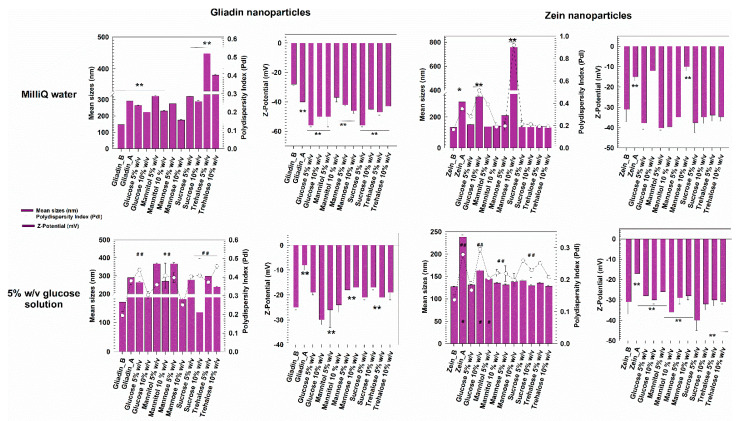
Mean diameter, size distribution, and Z-potential of freeze-dried protein-based nanoparticles prepared and redispersed in MilliQ water or 5% *w*/*v* glucose solution as a function of the cryoprotectant used. The DLS analyses were performed in MilliQ water. Gliadin/Zein_B = Mean size and polydispersity index before the lyophilization procedure. Gliadin/Zein_A = Mean size and polydispersity index after the lyophilization procedure. * *p* < 0.05; ** *p* < 0.001 with respect to the formulation prepared in MilliQ water before the lyophilization procedure. ^#^
*p* < 0.05; ^##^
*p* < 0.001 with respect to the formulation prepared in glucose solution before the lyophilization procedure.

**Figure 5 pharmaceutics-14-00332-f005:**
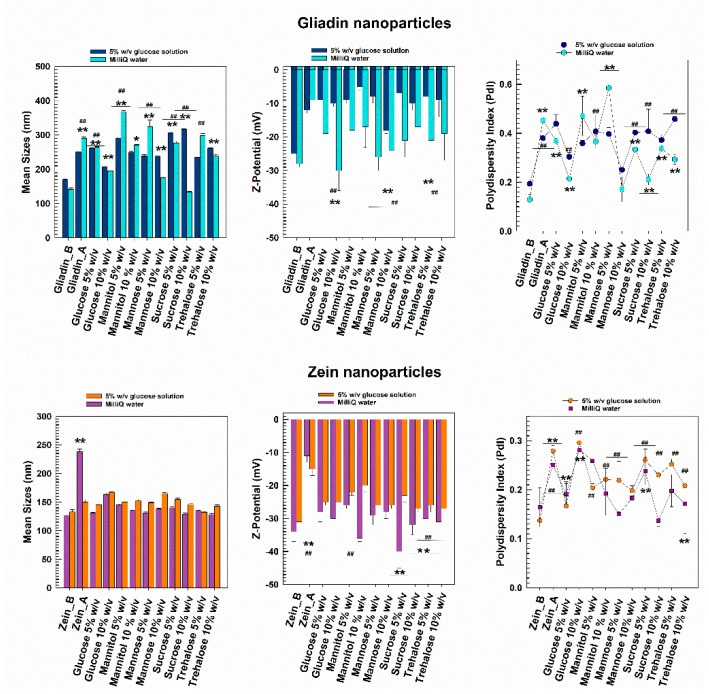
Influence of the dispersing medium used to perform the DLS analysis on the physico-chemical features of protein-based nanoparticles prepared in 5% *w*/*v* glucose solution as a function of the cryoprotectant used during the freeze-drying procedure. DLS analysis was performed at 25 °C in 5% *w*/*v* glucose solution or MilliQ water. Gliadin/Zein_B = Mean size and polydispersity index before the lyophilization procedure. Gliadin/Zein_A = Mean size and polydispersity index after the lyophilization procedure. * *p* < 0.05; ** *p* < 0.001 with respect to the formulation prepared in MilliQ water before the lyophilization procedure. ^##^
*p* < 0.001 with respect to the formulation prepared in glucose solution before the lyophilization procedure n.

**Figure 6 pharmaceutics-14-00332-f006:**
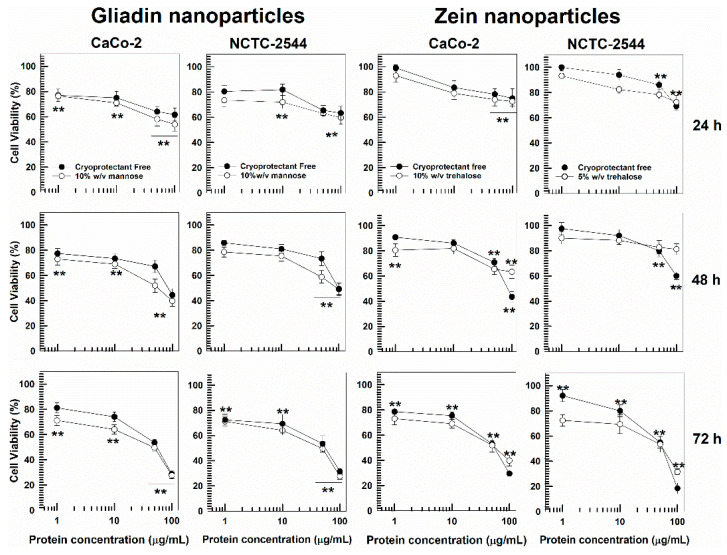
In vitro cytotoxicity of freeze-dried cryoprotectant-free and cryoprotectant-enriched zein- and gliadin-based nanoparticles as a function of protein concentrations and incubation times. Results are the average of three different experiments ± the standard deviation. ** *p <* 0.001 with respect to the control.

**Figure 7 pharmaceutics-14-00332-f007:**
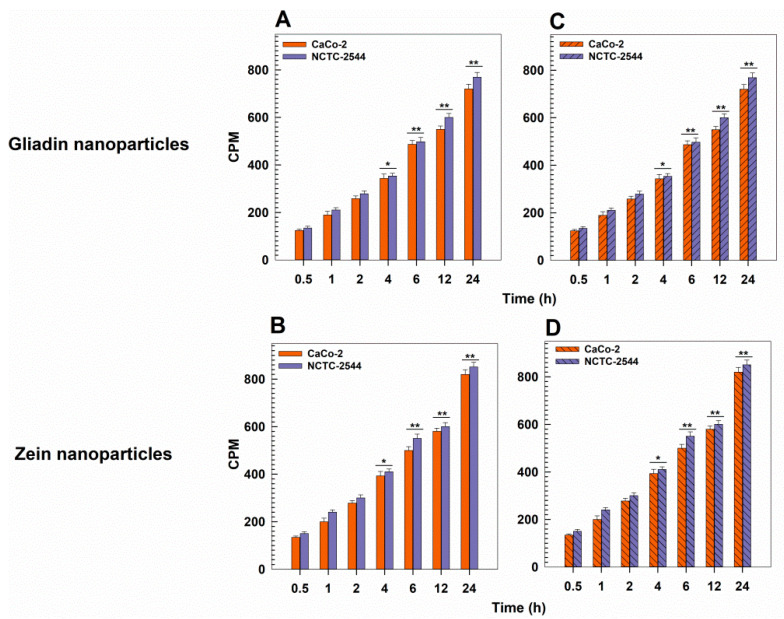
Interactions between tritiated gliadin and zein nanosystems and various human cell lines as a function of incubation times. (**A**): Freeze-dried cryoprotectant-free gliadin nanoparticles prepared with 1 mg/mL of polymer and 0.1% *w*/*v* of SRO2. (**B**): Freeze-dried cryoprotectant-free zein nanoparticles prepared with 2 mg/mL of polymer and 1.25% *w*/*v* of SD. (**C**): Freeze-dried gliadin nanoparticles containing 10% *w*/*v* of mannose as cryoprotectant. (**D**): Freeze-dried zein nanoparticles containing 10% *w*/*v* trehalose as cryoprotectant. Results are the mean of three different experiments ± standard deviation. * *p <* 0.05, ** *p <* 0.001 (with respect to the untreated cells).

**Table 1 pharmaceutics-14-00332-t001:** Mean diameter, size distribution, and surface charge of zein- and gliadin-based nanoparticles prepared in different polar media.

Composition of the Aqueous Phase	Mean Sizes (nm)	Polydispersity Index (PdI)	Z-Potential (mV)
Zein ^a^			
H_2_O (MilliQ)	133 ± 1	0.164 ± 0.007	−34 ± 3
Saline solution 0.9% *w*/*v*	472 ± 243 **	0.509 ± 0.22 **	−7 ± 1
Glucose solution 5% *w*/*v*	127 ± 1 *	0.137 ± 0.04	−31 ± 1
PBS 0.01 M pH 7.4	>1000 **	0.870 ± 0.170 **	−10 ± 2
Gliadin ^b^			
H_2_O (MilliQ)	143 ± 3	0.128 ± 0.01	−28 ± 2
Saline solution 0.9% *w*/*v*	794 ± 10 **	0.512 ± 0.03 **	−8 ± 1
Glucose solution 5% *w*/*v*	169 ± 2 *	0.194 ± 0.01 *	−22 ± 2
PBS 0.01 M pH 7.4	784 ± 35 **	0.680 ± 0.04 **	−10 ± 3

^a^ Zein-based nanoparticles prepared with 2 mg/mL of protein and 1.25% *w*/*v* of SD. ^b^ Gliadin-based nanoparticles prepared with 1 mg/mL of protein and 0.1% *w*/*v* of SRBO2. * *p* < 0.05; ** *p* < 0.001 with respect to the formulation prepared in MilliQ water. The DLS analysis was performed in the same medium used for sample preparation.

**Table 2 pharmaceutics-14-00332-t002:** Mean diameter, size distribution, and surface charge of zein- and gliadin-based nanoparticles prepared in MilliQ water and glucose solution analyzed in different polar media.

**Nanoparticles Prepared in MilliQ Water**						
**Medium of Analysis**	** *Zein Nanoparticles* ^a^ **			** *Gliadin Nanoparticles* ^b^ **		
	**Mean Sizes (nm)**	**polydispersity Index (PdI)**	**Z-Potential (mV)**	**Mean Sizes (nm)**	**Polydispersity Index (PdI)**	**Z-Potential (mV)**
H_2_O (MilliQ)	133 ± 1	0.164 ± 0.010	−34 ± 2	143 ± 3	0.128 ± 0.021	−28 ± 2
Saline solution 0.9% *w*/*v*	232 ± 27 **	0.187 ± 0.010	−13 ± 1	200 ± 12 **	0.258 ± 0.032	−7 ± 4
Glucose solution 5% *w*/*v*	161 ± 7 *	0.236 ± 0.04	−31 ± 1	183 ± 2 *	0.240 ± 0.025	−25 ± 2
PBS 0.01 M pH 7.4	180 ± 2 *	0.301 ± 0.037	−11 ± 1	465 ± 82 **	0.378 ± 0.035	−5 ± 1
**Nanoparticles Prepared in Glucose Solution (5% *w*/*v*)**						
**Medium of Analysis**	** *Zein Nanoparticles* ^a^ **			** *Gliadin Nanoparticles* ^b^ **		
	**Mean Sizes (nm)**	**Polydispersity Index (PdI)**	**Z-Potential (mV)**	**Mean Sizes (nm)**	**Polydispersity Index (PdI)**	**Z-Potential (mV)**
H_2_O (MilliQ)	132 ± 1 ^#^	0.144 ± 0.013	−33 ± 1	221 ± 1	0.218 ± 0.006	−25 ± 2
Saline solution 0.9% *w*/*v*	526 ± 123 ^##^	0.154 ± 0.019	−10 ± 1	445 ± 40 ^##^	0.193 ± 0.028	−8 ± 1
Glucose solution 5% *w*/*v*	127 ± 1 *	0.137 ± 0.04	−31 ± 3	169 ± 2	0.194 ± 0.01	−22 ± 1
PBS 0.01 M pH 7.4	257 ± 40 ^##^	0.187 ± 0.038	−12 ± 2	337 ± 56 ^##^	0.246 ± 0.021	−6 ± 3

^a^ Zein-based nanoparticles prepared with 2 mg/mL of biopolymer and 1.25% *w*/*v* of SD. ^b^ Gliadin-based nanoparticles prepared with 1 mg/mL of biopolymer and 0.1% *w*/*v* of SRBO2. * *p* < 0.05; ** *p* < 0.001 with respect to the formulation prepared in MilliQ water. ^#^
*p* < 0.05; ^##^
*p* < 0.001 with respect to the formulation prepared in glucose solution.

**Table 3 pharmaceutics-14-00332-t003:** Redispersibility Index (RDI) of gliadin- and zein-based nanoparticles prepared in different polar media.

	Cryoprotectant Concentration (% *w*/*v*)	RDI %		Cryoprotectant Concentration (% *w*/*v*)	RDI %
*Gliadin* (*MilliQ water*) ^a^	-	198 ± 10	*Zein* (*MilliQ water*) ^b^	-	224 ± 11
	Glucose 5%	180 ± 9		Glucose 5%	101 ± 5
	Glucose 10%	152 ± 8		Glucose 10%	249 ± 12
	Mannose 5%	187 ± 9		Mannose 5%	103 ± 5
	Mannose 10%	119 ± 6		Mannose 10%	570 ± 28
	Mannitol 5%	218 ± 11		Mannitol 5%	100 ± 5
	Mannitol 10%	156 ± 8		Mannitol 10%	108 ± 5
	Sucrose 5%	217 ± 11		Sucrose 5%	102 ± 5
	Sucrose 10%	196 ± 10		Sucrose 10%	101 ± 5
	Trehalose 5%	305 ± 15		Trehalose 5%	104 ± 2
	Trehalose 10%	259 ± 16		Trehalose 10%	99 ± 5
*Gliadin* (*5% w/v glucose solution*) ^a^	-	171 ± 9	*Zein* (*5% w/v glucose solution*) ^b^	-	187 ± 9
	Glucose 5%	156 ± 8		Glucose 5%	103 ± 5
	Glucose 10%	115 ± 6		Glucose 10%	128 ± 6
	Mannose 5%	217 ± 11		Mannose 5%	114 ± 6
	Mannose 10%	106 ± 5		Mannose 10%	106 ± 5
	Mannitol 5%	192 ± 10		Mannitol 5%	103 ± 5
	Mannitol 10%	160 ± 8		Mannitol 10%	109 ± 5
	Sucrose 5%	164 ± 8		Sucrose 5%	110 ± 6
	Sucrose 10%	79 ± 4		Sucrose 10%	102 ± 5
	Trehalose 5%	176 ± 9		Trehalose 5%	106 ± 5
	Trehalose 10%	141 ± 7		Trehalose 10%	101 ± 5

^a^ Gliadin-based nanoparticles prepared with 1 mg/mL of biopolymer and 0.1% *w*/*v* of SRBO2. ^b^ Zein-based nanoparticles prepared with 2 mg/mL of biopolymer and 1.25% *w*/*v* of SD.

## Data Availability

All data available are reported in the article.

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
