# Peer review of "Influence of the Dispersion Medium and Cryoprotectants on the Physico-Chemical Features of Gliadin- and Zein-Based Nanoparticles"

_pharmaceutics, 2022, doi:10.3390/pharmaceutics14020332_

Round 1

Reviewer 1 Report

The author has carefully revised the manuscirpt according to the reviewer's comments. It is ok. Now I recommend it accept for publication. 

Author Response

The authors are very grateful to the Reviewer for the positive judgment.

Reviewer 2 Report

In the paper "Influence of the dispersion medium and cryoprotectants on the physico-chemical features of gliadin-and zein-based nanoparticles", the authors synthesize and characterize, before and after lyophilization, gliadin- and zein-nanoparticles with different additives/excipients to increase the stability of a solid formulation. The subject is interesting, and the paper is overall well-written. However, several significant points need to be addressed before this manuscript may be considered for publication:

Materials and Methods:

In 2.2. the authors should clarify the procedure of nanoparticle production. Moreover, in this section authors described the production of nanoparticles using 3.3mg/ml of zein and 1.66 mg/ml of gliadin, according to the literature (lines 87-93). Despite this, in section 3.1 (Table 1, Table 2, and Table 3), the author's identify different concentrations. Please verify and clarify this difference.

Why did the authors select the turbiscan stability index and not the assessment by DLS of the samples during a predetermined period for the stability evaluation? How can this methodology assure the stability of nanoparticles after being resuspended to be used as nanocarriers in formulations as proposed by authors? This is of utmost relevance, and in my opinion, one of the major constraints of the work performed to be considered for publication.

Results and discussion:

All formulations showed an increase in size and PDI after lyophilization; the impact of these changes on particles properties should be better described and commented.

Author Response

The authors are very grateful to the Reviewer for the valued queries and advice. It is the opinion of the authors that the following changes, already reported in the previous version of the manuscript (the three concerns of the Reviewer are the same of those required in the last round of the revision), have improved the quality of the paper. A response to each point, raised in the main text, has been shown in red.

Reviewer 2 (R2): In 2.2. the authors should clarify the procedure of nanoparticle production. Moreover, in this section authors described the production of nanoparticles using 3.3 mg/ml of zein and 1.66 mg/ml of gliadin, according to the literature (lines 87-93). Despite this, in section 3.1 (Table 1, Table 2, and Table 3), the author's identify different concentrations. Please verify and clarify this difference.

A: In response to the Reviewer’s comment, the nanoprecipitation technique used for the development of zein- and gliadin-based nanoparticles, is based on the association of two miscible solvents, one of them being a good solvent (ethanol), and the other one acting as a non-solvent (water). When these two phases are mixed, a supersaturation of the material forming the particles occurs as the organic solvent migrates toward the aqueous phase, and this promotes the aggregation of the polymer in the form of nanoparticles.

In this regard, in section 2.2 the amount of each biomaterial used was expressed taking into consideration the volume of organic phase used for the solubilization of each polymer (3 mL of an ethanol/water mixture). Successively, the concentration of each component was expressed with respect to the volume of formulation obtained after the evaporation of the organic solvent (ethanol), which was equal to 5 mL. Therefore, the final amount of polymer used was 2 mg/mL for zein and 1 mg/mL for gliadin.

However, following the Reviewer’s suggestion and in order to avoid any misleading of the reader, the amount of zein and gliadin used in the preparation procedure has been duly unified and section 2.2 was modified (lines 146-147).

R2: Why did the authors select the turbiscan stability index and not the assessment by DLS of the samples during a predetermined period for the stability evaluation? How can this methodology assure the stability of nanoparticles after being resuspended to be used as nanocarriers in formulations as proposed by authors? This is of utmost relevance, and in my opinion, one of the major constraints of the work performed to be considered for publication.

A: The authors thank the Reviewer for the interesting question.

It is the opinion of the authors that a brief description of the theory governing Multiple Light Scattering and Turbiscan Technology would be useful towards providing a conceivable answer.

As reported in the main text (lines 163-174), Turbiscan® technology is based on the Static Multiple Light Scattering principle. An infrared light source (wavelength λ=880 nm) illuminates the sample and two sensors collect the Backscattered (BS) and Transmitted (T) signals. The BS and T signals are acquired repeatedly over time throughout the entire height of the sample. The composition of these scans allows the detection of physical instabilities present in the dispersion such as aggregation, sedimentation or creaming. In fact, any destabilization phenomenon going on will necessarily have an impact on the BackScattering and/or Transmission signal intensities during the aging process. Each phenomenon can be detected and quantified based on BS and/or T signal intensities. The TSI calculation is based on an integrated algorithm that sums up the evolution of T or BS light at every position measured (h), based on a scan-to-scan difference, over the total sample height (H):

where t_max is the measurement point corresponding to the time t at which the ??? is calculated, z_min and z_max the lower and upper selected height limits respectively, N_h = (z_max - z_min)/Δh the number of height positions in the selected zone of the scan and ?ST the considered signal (?? if ? < 0.2%, ? otherwise). The proposed algorithm generates a curve with a certain angle/slope due to the parameters used.

TSI also corresponds to a cumulative sum of all the BS or T variations of the full sample (the Turbiscan apparatus allows the analysis of both transparent and opaque samples). As the scans add up over time the TSI value can change and at a given time a single TSI value is associated and the “Destabilization kinetics” graph can be created showing the TSI evolution over time.

The TSI was developed in order to easily and simultaneously compare and characterize the physical stability of multiple formulations. There is no established range of TSI values for which a formulation can be considered stable. It depends on the analyzed sample. In general, the higher the TSI value, the less stable the sample. Lower TSI values indicate more stable samples  as compared to formulations with higher TSI values. For this reason, the BS and T profiles of the discussed formulations have been included in the supplementary materials.

The aim of the authors was to evaluate the influence of the medium on the stability of the various protein nanoparticles (new Figures 2 and 3).

The Turbiscan apparatus is recognized as a valid tool for analyzing the stability of samples without dilution or mechanical stress. It is employed by many research groups for the analysis of different samples: Gagliardi et al., 2019, Heliyon, 5, e02422. https://10.1016/j.heliyon.2019.e02422; Cosco et al., 2019, Int. J. Biol. Macromol., 132, 550-557. https://doi.org/10.1016/j.ijbiomac.2019.03.241; Rezvani et al., 2019, Pharmaceutics, 11, 13. https://doi.org/10.3390/pharmaceutics11010013; Du and Wang, 2020, Molecules, 25, 393.  https://doi.org/10.3390/molecules25020393; Puglia et al., 2020, Nanomaterials, 10, 287. https://doi.org/10.3390/nano10020287; Yuan et al., 2020, Food Chemistry, 126709. https://doi.org/10.1016/j.foodchem.2020.126709.

The authors have reported only a few examples of the most recent articles that have employed this technology. In the abovementioned works it is possible to observe how the evaluation of the TSI varies according to the analyzed sample and it is also possible to see lines characterized by a specific slope (more or less evident based on the stability of the sample).

The authors hope that this explanation has fully satisfied the Reviewer’s perplexity.

R2: All formulations showed an increase in size and PDI after lyophilization; the impact of these changes on particles properties should be better described and commented. For example, in lines 282-293 and 294-301.

A: According to the Reviewer’s comment, the discussion was revised (lines 400-407, 415-418 and 421-423).

This manuscript is a resubmission of an earlier submission. The following is a list of the peer review reports and author responses from that submission.

Round 1

Reviewer 1 Report

In this paper, the authors discuss the impact of various media (0.9% w/v NaCl, 5% w/v glucose and phosphate buffered saline solution) on the characteristics of nanosystems made up of two vegetal proteins, and their physico-chemical properties were analyzed after the freeze-drying procedure. Although their results are interesting, there are a number of issues for the authors to consider for the improvement.

Majors:

  1. In the abstract,the author mentioned that “Mannose emerged as a useful excipient for the gliadin nanosystems prepared in water, whereas the use of sucrose, glucose and mannose in the development of freeze-dried systems conferred long-term storage characteristics to them when a glucose solution was used in their preparation”, why the mannose appears twice? And it is not clear exactly which one has been choose to used as cryoprotectant.
  2. In the abstract, the purpose of this is experiment is not clear enough, and the logic is confusing. The part should be condensed.
  3. In the “Freeze-drying of zein and gliadin nanoparticles” part, the Freeze-drying experimental conditions are not described for this experiment.
  4. As author mentioned earlier, “it would be useful to investigate their behavior not only in their “original media” (i.e., in the solvent used during the preparation procedure), but also in physiological media in order to evaluate their performance once administered in the body”, but the result proved that the mean sizes of the samples increased in the PBS (7.4), but the pH value of blood is 7.4, how did the authors solve the problem that the mean sizes increase in the blood after intravenous injection administration?
  5. In page 7, the results proved that the worst results were obtained using trehalose as freeze-drying protectant, but lots of literature stated that trehalose is an excellent freeze-drying protectant, why the experiments come to opposite conclusions?

Minors:

  1. In page 3, No centrifugation experimental conditions are described for this experiment.
  2. In the Figure 2, it is recommended that the authors mark the data in the figure, make it easier for readers to follow the figures and data.

Reviewer 2 Report

In this manuscript authors have investigated effect of dispersion medium and cryoprotectant on size of protein-based nanoparticles. After carefully reviewing this manuscript, I recommend publication of this manuscript in the journal after revision for following points:

The major concern in this research article is that authors have investigated effect of dispersion medium as well as of cryoprotectant on size and PDI of zein and gliadin protein-based nanoparticles, hence the title of manuscript is required to be revised accordingly. Further in introduction authors discussed about cryoprotectant. Discussion about dispersion medium is also required in significant amount in the introduction. 

Abbreviations should have been written carefully. In introduction they have used SMLS and SLMS for static multiple light scattering in introduction at line no 59 and 69 on page no 2

Further author must have also investigated effect on zeta potential also along with size, as zeta potential is an important criterion for deciding stability of colloidal carrier systems. 

Further, SEM/AFM/TEM images of nanoparticles are also required to be provided in the manuscript. 

Few corrections are required for typographical, grammatical error or for improper usage of English in abstract and introduction. Few examples are given below:

Page no. 1: Line 11-14

Page no. 1: line 15: demonstrated in place of demonstrate

Page no. 1: Line 19-21

Reviewer 3 Report

In the paper "Influence of the dispersion medium on the mean diameter of gliadin- and zein-based nanoparticles", the authors synthesize and characterize, before and after lyophilization, gliadin- and zein-nanoparticles with different additives/excipients to increase the stability of a solid formulation. The subject is interesting, and the paper is overall well-written. However, several significant points need to be addressed before this manuscript may be considered for publication:

General comments: Please confirm the use of abbreviates and the tables and figures numbers (in the tables, figures and the text.)

Materials and Methods:

In 2.2. the authors should clarify the procedure of nanoparticle production. Moreover, in this section authors described the production of nanoparticles using 3.3mg/ml of zein and 1.66 mg/ml of gliadin, according to the literature (lines 87-93). Despite this, in section 3.1 (Table 1, Table 2, and Table 3), the author's identify different concentrations. Please verify and clarify this difference.

Also, in this section is not clear the process of centrifugation. Why author's perform this centrifugation step? Also, did the nanoparticles have been recovered and resuspended in different media?

Why did the authors select the turbiscan stability index and not the assessment by DLS of the samples during a predetermined period for the stability evaluation? How can this methodology assure the stability of nanoparticles after being resuspended to be used as nanocarriers in formulations as proposed by authors? This is of utmost relevance, and in my opinion, one of the major constraints of the work performed to be considered for publication.

Why did the authors not evaluate the different nanoparticles and suspensions' zeta potential once the main data presented was referred to DLS analysis?

The physicochemical characterization of nanoparticles should include other important parameters, such as morphology and why biocompatibility and cell internalization were not tested. Despite the previous works describing gliadin- and zein- nanoparticles, the effect of additives on cytotoxicity and cell interaction and internalization may be relevant.

Results and discussion:

In lines145-152, the authors referred to previous works with concentrations of gliadin and zein different from the concentrations used in this work. Why?

If the nanoparticles were prepared under the same conditions, why data presented in Table 1 and Table 2 are different? (e.g. first lines of each Table).

All formulations showed an increase in size and PDI after lyophilization; the impact of these changes on particles properties should be better described and commented. For example, in lines 282-293 and 294-301.